# The Effect of Vitamin K2 Supplementation on PIVKA-II Levels in Patients with Severe Motor and Intellectual Disabilities Undergoing Long-Term Tube Feeding

**DOI:** 10.3390/nu15214525

**Published:** 2023-10-25

**Authors:** Hiromitsu Ohmori, Akihiko Kato, Yuka Shirai, Reiji Fukano, Akiko Nagae, Masami Yamasaki, Junko Komenaka, Eiji Imamura, Masao Kumode, Takafumi Miyachi

**Affiliations:** 1Department of Pediatrics, National Hospital Organization Yanai Medical Center, 95 Ihonosho, Yanai 742-1352, Japan; 2Blood Purification Unit, Hamamatsu University Hospital, 1-20-1 Handayama, Higashi-ku, Hamamatsu 431-3192, Japan; a.kato@hama-med.ac.jp; 3Clinical Nutrition Unit, Hamamatsu University Hospital, 1-20-1 Handayama, Higashi-ku, Hamamatsu 431-3192, Japan; yu-shira@hama-med.ac.jp; 4Department of Pediatrics, Yamaguchi University Graduate School of Medicine, 1-1-1 Minami-Kogushi, Ube 755-8505, Japan; fukano.r@yamaguchi-u.ac.jp; 5Department of Pediatrics, Biwako Gakuen Kusatsu Medical and Welfare Center for Children and Persons with Severe Motor and Intellectual Disabilities, 8-3-113 Kasayama, Kusatsu 525-0072, Japan; a_nagae@biwakogakuen.or.jp (A.N.); m_kumode@biwakogakuen.or.jp (M.K.); 6Department of Neurology, National Hospital Organization Yanai Medical Center, 95 Ihonosho, Yanai 742-1352, Japan; yamasaki.masami.uk@mail.hosp.go.jp (M.Y.); eijiima5@yahoo.co.jp (E.I.); miyachi.takafumi.ky@mail.hosp.go.jp (T.M.); 7Department of Clinical Nutrition, National Hospital Organization Yanai Medical Center, 95 Ihonosho, Yanai 742-1352, Japan; komenaka.junko.ky@mail.hosp.go.jp

**Keywords:** severe motor and intellectual disabilities, enteral nutrition, vitamin K, protein induced by vitamin K absence or antagonists (PIVKA)-II, undercarboxylated osteocalcin (ucOC)

## Abstract

Nutritional support is essential for patients with severe motor and intellectual disabilities (SMID) to ensure the smooth provision of medical care. These patients often require long-term tube feeding with enteral formulas, potentially leading to deficiencies in vitamins and trace elements. Additionally, frequent antibiotic use for infections often disrupts gut microbiota, inhibiting vitamin K2 production by intestinal bacteria. We assessed the serum protein induced by vitamin K absence or antagonists-II (PIVKA-II) and undercarboxylated osteocalcin (ucOC) levels to assess the vitamin K status in 20 patients with SMID (median age: 44.1 years, 11 men and 9 women) undergoing long-term tube feeding for durations ranging from 3 to 31 years. Thirteen (65%) and nine (45%) patients had elevated PIVKA-II (<40 mAU/mL) and serum ucOC levels (reference value < 4.50 ng/mL), respectively. Dietary vitamin K1 intake did not differ between patients with and without elevated PIVKA-II levels. Vitamin K2 supplementation for 3 months decreased serum PIVKA-II levels near those within the reference range. Approximately half of the patients with SMID on tube feeding had subclinical vitamin K deficiency. Further studies are needed to ascertain if long-term vitamin K2 supplementation effectively prevents vitamin K deficiency-induced hypercoagulation, osteoporosis, and vascular calcification in patients with SMID.

## 1. Introduction

Severe motor and intellectual disabilities (SMID) are characterized by severe physical and mental retardation. Japan has approximately 40,000–50,000 patients with SMID [1,2,3]. Most patients with SMID are bedridden or remain in a sitting position and have intelligence quotients (IQ) lower than 35 (Ohshima’s classification, Grade 1–4) (Appendix A) [4].

SMID are caused by various diseases such as cerebral palsy, cerebral injury, congenital anomalies, chromosomal aberrations, and meningitis and/or encephalitis during the developmental stages in the perinatal period and infancy, which severely damage the central nervous system [3,4,5]. Moreover, SMID are frequently complicated by refractory epilepsy [2]. 

Nutritional support is critical to ensure the smooth provision of medical care on par with that provided for the respiratory, circulatory, and vascular systems [6]. Patients with SMID often experience feeding problems, such as difficulty in swallowing due to advanced oral motor dysfunction [1,2,3,4,5,6,7]. Especially, children with SMID who need respiratory assistance are at high risk of malnutrition [7], often experiencing gastroesophageal reflux (GER), which causes reflux esophagitis, esophageal hemorrhage, and aspiration pneumonia [1,7]. Therefore, percutaneous endoscopic gastrostomy (PEG) and surgical fundoplication are performed to maintain artificial enteral feeding [8,9]. PEG with jejunal extension (PEG-J) has also been used to mitigate GER symptoms [10,11]. However, commercially available enteral formulas do not aim for long-term administration; thus, the total amounts of some micronutrients are insufficient in patients with SMID undergoing long-term PEG or PEG-J. 

Vitamin K is a fat-soluble vitamin that plays an important role in blood coagulation, bone mineral metabolism, and vascular calcification. It acts as a cofactor to convert the glutamyl residue of the coagulation factors (II, VII, IX, and X) to the γ-carboxyglutamyl residue. Vitamin K also induces the synthesis of osteocalcin (OC), the most abundant non-collagenous bone matrix protein, and regulates bone mineralization. 

In patients with SMID, osteoporosis starts during the growth period, especially in those with tube feeding [12]. A systematic review [13] demonstrated that incident bone fractures are observed in approximately 4% of patients per year, whereas the prevalence of low bone mineral density (BMD) in the femur was 77% in children with severe cerebral palsy. Long-term tracheostomy results in granulomatous tracheomalacia and trachea innominate artery fistulas, eventually leading to a potential risk of massive tracheal bleeding [14]. Thus, vitamin K deficiency could advance bone mineral disorders and bleeding tendencies. However, the characteristics of vitamin K deficiency in patients with SMID remain unclear.

This study aimed to assess vitamin K deficiency in patients with SMID who were undergoing long-term tube feeding. We first measured the serum protein induced by vitamin K absence or antagonism for factor II (PIVKA-II) and the total OC and undercarboxylated fraction of OC (ucOC) as markers of vitamin K status in the liver and bones [15]. Next, we examined the effect of a 3-month treatment with a vitamin K2 preparation, menaquinone (MK)-4, a major form of vitamin K in the tissue, on serum PIVKA-II levels in patients with elevated PIVKA-II levels. 

## 2. Materials and Methods

### 2.1. Study Design

Among the 75 institutionalized patients with SMID in our hospital, categorized as class 1 or 2 according to Ohshima’s classification, we excluded 37 patients who were capable of self-feeding. Moreover, we excluded 17 patients who simultaneously took a meal orally and received enteral formulas using PEG or PEG-J tube. Additionally, we excluded one patient with hepatocellular cancer. Finally, 20 patients on tube feeding nutrition were eligible for this study.

### 2.2. Blood Sampling and Laboratory Examinations

Blood samples were collected from each patient during regular laboratory examinations for complete blood count, biochemistry, and rapid turnover protein analyses (abbreviations are listed in Appendix A).

Furthermore, we measured serum PIVKA-II (reference range, <40 mAU/mL), OC (reference range, male; 8.4–331.1 ng/mL, premenopausal female; 7.8–30.8 ng/mL, menopausal female; 14.2–54.8 ng/mL), and ucOC (reference range, <4.50 ng/mL) levels using an electrochemiluminescence immunoassay (BML, Biomedical Laboratory, Co., Ltd., Tokyo, Japan). 

### 2.3. Assessment of Dietary Vitamin K Intake

The daily intake of macro- and micronutrients was calculated based on the ingredient lists of enteral formulas and micromineral supplementation (Appendix A).

We calculated the percentiles of daily energy and vitamin K intakes according to the Dietary Reference Intakes for the Japanese (2020) (Appendix A) [16].

### 2.4. Supplementation of Vitamin K

A prior dose-finding study of menatetrenone established that a daily dose of 45 mg serves as the minimum effective dose for enhancing bone cycle parameters in Japanese postmenopausal women with osteoporosis [17]. Consequently, a daily dose of 45 mg is pharmacologically designated for osteoporosis treatment in Japan. Thus, we opted for three capsules of the vitamin K2 preparation, MK-4 (menatetrenone capsule, 15 mg, Towa Pharmaceutical Co., Ltd., Osaka, Japan), in adult patients with PIVKA-II levels exceeding 100 mAU/mL. Given that the safety of MK-4 has yet to be confirmed in pediatric subjects, the daily dose was reduced to 30 mg for patients younger than 15 years. We administered MK-4 for a duration of three months, as this time frame has been shown to be safe without inducing any thrombotic tendencies in Japanese elderly patients with osteoporosis [18].

### 2.5. Assessment of Bone Quality at the Calcaneus

We measured BMD T-score (the number of standard deviation (SD) below young adult mean) and Z-score (the number of SD below age- and gender-matched average) at the right calcaneus by quantitative ultrasonography (GE Healthcare Achilles, Horten, Norway) before and after MK-4 treatment in patients who can hold a sitting position at measurement. 

### 2.6. Statistical Analyses

Statistical analyses were performed using the SPSS version 18.0 software for Windows (SPSS Inc., Tokyo, Japan). Data between the two groups were compared using the Mann–Whitney test. A multiple regression analysis was performed to identify the independent variables that affect PIVKA-II levels. Asterisks denote *p*-values less than 0.05. The correlation between the percentile of daily vitamin intake and the ucOC/OC ratio was analyzed using Spearman’s rank test. Data are expressed as mean ± SD. 

### 2.7. Ethics Statement

Detailed medical information about the present study was provided to the subjects or proxies, and written consent was obtained from the guardians of patients with limited capacity to provide consent for participation in the study. This study was approved by the Regional Ethical Review Board of the National Hospital Organization Yanai Medical Center (Y-4-9) and was conducted in accordance with the World Medical Association’s Declaration of Helsinki guidelines. Patients with SMID were admitted to the recuperation and long-term care wards at the Yanai Medical Center. Subsequently, we conducted our research using anonymous clinical data under close supervision after approval by the medical ethics committee of our hospital.

## 3. Results

### 3.1. Patients’ Characteristics

The patients comprised 11 men and nine women with a median age of 44.1 (9–70 years). The median body weight (BW) was 39.9 kg, ranging from 28.4 to 62.5 kg. Body mass index (BMI) was 18.4 ± 4.7 kg/m^2^, ranging from 13.3 to 30.7 kg/m^2^. Seventeen patients were categorized as having Grade 1 (bedridden state, IQ < 20), and the other three patients were categorized as having Grade 2 (sitting position, IQ < 20) using Ohshima’s classification of SMID [4]. The causes of SMID were cerebral palsy (*n* = 11), chromosomal abnormalities (*n* = 3), mental retardation (*n* = 3), sequelae of measles encephalitis (*n* = 1), neurodegenerative disease (*n* = 1), and congenital anomalies (*n* = 1). 

Eleven patients underwent tracheostomy, and 10 required mechanical ventilation support. Two patients received nasal noninvasive positive pressure ventilation. Sixteen patients had refractory epilepsy and were treated with antiepileptic agents. All the patients had complications of advanced spinal deformity. 

Prophylactic antibacterial therapy for recurrent pulmonary and urinary tract infections was administered to 19 patients (95%). The antibiotics used included erythromycin (*n* = 18), sulfamethoxazole/trimethoprim (*n* = 17), clarithromycin (*n* = 1), and meropenem (*n* = 1). 

Six patients had previously experienced bone fractures, including humeral (*n* = 2), femoral (*n* = 2), pelvic (*n* = 1), and tibial (*n*= 1) fractures. None of the patients experienced bleeding events such as gastrointestinal or tracheal bleeding.

### 3.2. Enteral Tube Feeding

All patients had been undergoing long-term tube feeding (nasogastric tube, *n* = 3; PEG or PEG-J, *n* = 17) with a mean duration of 12.3 (3–31) years. The prescribed enteral formulas were as follows: Clinimeal^®^ (*n* = 13), Elental P^®^ (*n* = 4), Meibalance RHP^®^ (*n* = 2), and Meiflow^®^ (*n* = 1) (Appendix A). Trace elements were also supplied as follows: Teson^®^ (*n* = 14), Teson^®^ + V CRESC CP10^®^ (*n* = 3), or V CRESC CP10^®^ alone (*n* = 2). 

The mean daily intake of energy was 960 ± 208 (620–1440) kcal/day, corresponding to 43.4% of the recommended daily intake of healthy subjects (Appendix A) [16]. The mean dietary protein, fat, and carbohydrate intakes were 45.5 ± 12.2 (26.3–70) g/day, 29.9 ± 29.5 (4.5–147.1) g/day, and 134.4 ± 32.6 (91.8–209.1) g/day, respectively. 

### 3.3. Vitamin K Status

The mean vitamin K intake was 59 ± 5 (21 to 112) µg/day, corresponding to 40.1 ± 12.8% of the recommended adequate intake (Appendix A) [16]. 

The median serum concentrations of PIVKA-II, OC, and ucOC were 409.8 ± 140.2 (21–2080) mAU/mL, 25.7 ± 11.3 (8.0–45.3) ng/mL, and 5.06 ± 0.85 (1.26–14.42) ng/mL, respectively. Serum PIVKA-II levels exceeded the reference value (<40 mAU/mL) in 13 of 20 patients (65%). Increased serum ucOC levels were also found in 9 patients (45%). 

We compared the clinical parameters between patients with SMID who had normal PIVKA-II levels (*n* = 7) and those with elevated PIVKA-II levels (*n* = 13), as detailed in Table 1.

Serum albumin was significantly lower in patients with increased PIVKA-II levels than in those with normal levels (3.6 ± 0.1 vs. 3.8 ± 0.2 g/dL, *p* < 0.05). Other biochemical nutritional parameters such as transthyretin, transferrin, and retinol-bonding protein were within normal references, and no difference was found between the two groups. Blood coagulation parameters were also identical between the two groups (Table 1). The daily vitamin K intake was lower in patients with increased PIVKA-II levels than in those with normal (53 ± 5 vs. 70 ± 9 μg/day) levels, but no statistically significant difference was found (*p* = 0.28). Dairy vitamin K intake adjusted for current BW (mean: 1.51 ± 0.58 μg/kg actual BW/day) was not an independent predictor of increased PIVKA-II following the multiple regression analysis adjusted for age, BMI, and duration of tube feeding. 

Furthermore, we compared the clinical parameters between patients with normal (*n* = 11) and elevated ucOC (*n* = 9) levels. There were no significant differences in any of the clinical parameters between the two groups (Table 2). The vitamin K intake did not correlate with the ucOC/OC ratio, a sensitive marker of bone vitamin K status (correlation coefficient = −0.22, *p* = 0.32). 

Three out of the 13 patients with elevated PIVKA-II levels had prolonged prothrombin times (PT) (>13.0 s). In particular, in one case where the PIVKA-II level was elevated to 1892 mAU/mL and uOC to 8.75 ng/mL, PT, PT-INR, and aPTT were increased to 30.9 s, 2.97, and 51.8 s, respectively. Elevated PIVKA-II levels were also observed in five of the six patients who had experienced bone fractures. 

### 3.4. Effect of Vitamin K Supplementation on PIVKA-II Levels

We administered 45 mg/day of MK-4 in 6 adult patients and 30 mg/day in 2 pediatric patients whose PIVKA-II levels exceeded 100 mIU/mL. Mean serum PIVKA-II levels dramatically decreased from 946 ± 718 (179–2080) to 35 ± 20 (12–46) mAU/mL (*p* < 0.01) (Figure 1). No patient had experienced incident cardiovascular events, including sudden death, during the treatment. 

### 3.5. Effect of Vitamin K Supplementation on BMD Parameters at the Calcaneus

We measured the BMD T-score and Z-score at the right calcaneus using quantitative ultrasonography in 10 adult patients (age: 54 ± 8 years old, male/female = 7/3). The heel T-score ranged from −4.1 to −6.1 with a mean value of −6.0 ± 1.0, which fulfilled the diagnostic criteria of osteoporosis of less than −2.5. Furthermore, Z-scores ranged from −2.4 to −6.1 (mean: −4.3 ± 1.1), lower than the criteria of osteoporosis (<−2.0) in all patients. 

Out of 10 patients, four adult patients received 45 mg of MK-4 daily for 3 months. However, there was no difference in T-score and Z-score between those before and after MK-4 treatment. Similarly, heel BMD T-score and Z-scores did not change in patients without MK-4 treatment during the same observation period. 

## 4. Discussion

Patients with SMIDs require tube feeding because of the high incidence of gastroesophageal reflux disease and dysphagia [1,5,7]. However, anthropometric assessment is difficult owing to excess body fluid, short stature, and severe scoliosis in patients with SMID. Thus, the management of energy intake from enteral feeding is based on changes in BW and biochemical data from periodic blood sampling at a low frequency. 

The resting energy expenditure (REE) is reported to be lower in patients with SMID than in healthy subjects because of the reduced amount of fat-free mass. Therefore, the existing weight-based REE equations may overestimate the REE compared with the measured REE. For example, the root-mean-squared error between the measured REE using indirect calorimetry and predicted REE from the Harris–Benedict equation was 185.2 kcal/day in patients with SMID aged over 18 years [19]. The doubly labeled water method, a gold standard for measuring total energy expenditure (TEE), revealed that the TEE was lowest in children aged 6–15 years with motor disabilities [20]. In this study, the mean daily energy and protein intakes were 24 kcal/kg/actual BW/day and 1.1 g/kg/actual BW/day, respectively. Despite the lower energy intake with respect to the recommended intake [16], the mean serum transthyretin level was approximately 30.0 mg/dL (Table 1), indicating that energy restriction (about 25 kcal/kg/day) does not cause clinically overt malnutrition in tube-fed patients with SMID in this study.

Commercially available enteral formulas are not intended for long-term administration; thus, the total amounts of some micronutrients are insufficient (Appendix A). In this study, we showed that the daily vitamin K1 intake was decreased to 58.5 ± 21.4 (21.2–112) µg/day due to the lower content of vitamin K1 in the prescribed enteral formulas (4.49 to 8.0 µg/100 kcal), which accounted for approximately 40% of the recommended intake (Appendix A) [16]. Prolonged treatment with certain antibiotics can also interfere with vitamin K2 production by normal intestinal flora. Anticonvulsant therapy (phenytoin) could be a potent risk factor for vitamin K deficiency in patients with SMID. 

In this study, we measured serum levels of PIVKA-II and ucOC as biomarkers of vitamin K deficiency. We found that serum PIVKA-II and ucOC were elevated in 65% and 45% of the patients, respectively. In contrast, prolonged PT was found only in 3 of 13 patients with high PIVKA-II levels. Thus, serum PIVKA-II is a more sensitive marker to detect vitamin K deficiency in the liver. In agreement with this finding, it was reported that PT did not correlate to dietary vitamin K1 intake in patients with SMID [1]. 

Several studies have examined the prevalence of vitamin K deficiency in patients with SMID [1,5,12,21]. Yoshikawa et al. [21] first reported that 9 of 21 children with SMID had vitamin K deficiency based on their serum PIVKA-II levels. They found that infection, antibiotic use, and elemental nutrition were the risk factors for vitamin K deficiency [21]. Similarly, Nagae et al. [1] reported that the serum levels of PIVKA-II and ucOC were above the upper reference range in 52% and 30% of patients, respectively. Vitamin K and D co-deficiencies have also been reported in patients with SMID [12]. 

Vitamin K2 is synthesized by bacteria in the intestine. In this study, a synthetic vitamin K2 (menatetrenone) supplementation (30 or 45 mg/day) for 3 months decreased the PIVKA-II levels near to the reference range (Figure 1). However, the treatment did not improve BMD T- and Z-scores at the calcaneus (Table 3). This negative effect of MK-4 on bone health may be mainly due to a shorter observation period. A previous study [22] demonstrated that 45 mg/day of MK-4 treatment for 48 weeks significantly increased bone formation markers such as bone alkali-phosphatase and OC, while there was no significant change in BMD at the second metacarpal bone by radio-absorptiometry after the treatment [22]. Thus, a longer treatment period of MK-4 over 6 months will be needed to improve osteoporosis in SMID patients. Alternatively, developing natto (fermented soybeans)-rich formulas containing an enzyme natto-kinase may be useful to supply MK-7.

Phylloquinone (vitamin K1) is the primary type of dietary vitamin K and is found abundantly in leafy green vegetables and vegetable oils. Vitamin K1 is thought to contribute to bone health equivalently to vitamin K2, as all vitamin K molecules are converted into MK-4, subsequently becoming active. A vitamin K1 depletion–repletion study demonstrated that 250 to 1000 μg/day of vitamin K1 is needed for reducing the serum ucOC level within 1 to 2 weeks [23]. In healthy volunteers who have maintained stable anticoagulation, food supplementation with 150 μg/day of vitamin K1 was sufficient to reduce elevated PT-INR, whereas more than 300 μg/day was needed to reduce the ucOC levels [24]. Cheung et al. [25] also reported that high-dose vitamin K1 supplementation (5 mg/day) for 2–4 years did not protect against age-related decline in BMD at the lumber spine, total hip, femoral neck, or ultra-distal radius; however, it prevented clinical fractures in postmenopausal women with osteopenia. These findings collectively suggest that vitamin K1 dosages higher than those recommended by the reference (Appendix A) are required to improve bone mineral metabolism to prevent blood coagulation abnormalities. This difference is probably due to the first pass effect, since vitamin K is absorbed from the intestine and first utilized in the liver, then transferred to the other organs, including bone.

In patients with SMID, Nagae et al. [1] demonstrated that the dietary vitamin K1 requirement for normal ucOC (<4.5 ng/mL) was greater than 5.5 μg/kg BW/day, while that for normal PIVKA-II (<28 mAU/mL) was above 2.5 μg/kg BW/day in those without antibiotics therapy. Furthermore, Kuwabara et al. [5] demonstrated that an increase in median vitamin K1 supply from 66 to 183 µg/day using a vitamin K1-rich enteral formula containing 62.5 μg/100 mL (Racol^®^, Otsuka Pharmaceutical Factory, Inc., Naruto, Japan) for 3 months decreased serum concentration of PIVKA-II but not that of ucOC [5]. However, commercially available enteral formulas currently contain only vitamin K1 less than 10 μg/100 mL (Appendix A). Therefore, a long-term supply of high-dose vitamin K1 is not clinically applicable in patients with SMID with tube feeding. 

This study has some limitations. First, the number of enrolled patients was small; however, it seems that this is an irrelevant problem to study. Secondly, we did not examine the effect of nutritional vitamin K1 supplementation at the recommended dose (Appendix A) on serum PIVKA-II levels. Third, we did not directly measure serum vitamin K1 and K2 levels. Lastly, although the causes of death in patients with SMID after 1990 were heart failure, other cardiovascular diseases, and sudden death, accounting for 13.0%, 4.7%, and 4.7% of cases, respectively [26], we did not evaluate the role of vitamin K deficiency in cardiovascular diseases including vascular calcification.

## 5. Conclusions

Vitamin K deficiency is highly prevalent in tube-fed patients with SMID. Therefore, monitoring blood PIVKA-II and ucOC levels may be useful for detecting subclinical vitamin K deficiency in such patients. Future studies are needed to determine whether long-term supplementation with vitamin K2 prevents vitamin K deficiency-induced bleeding, osteoporosis, and arteriosclerosis. 

## Figures and Tables

**Figure 1 nutrients-15-04525-f001:**
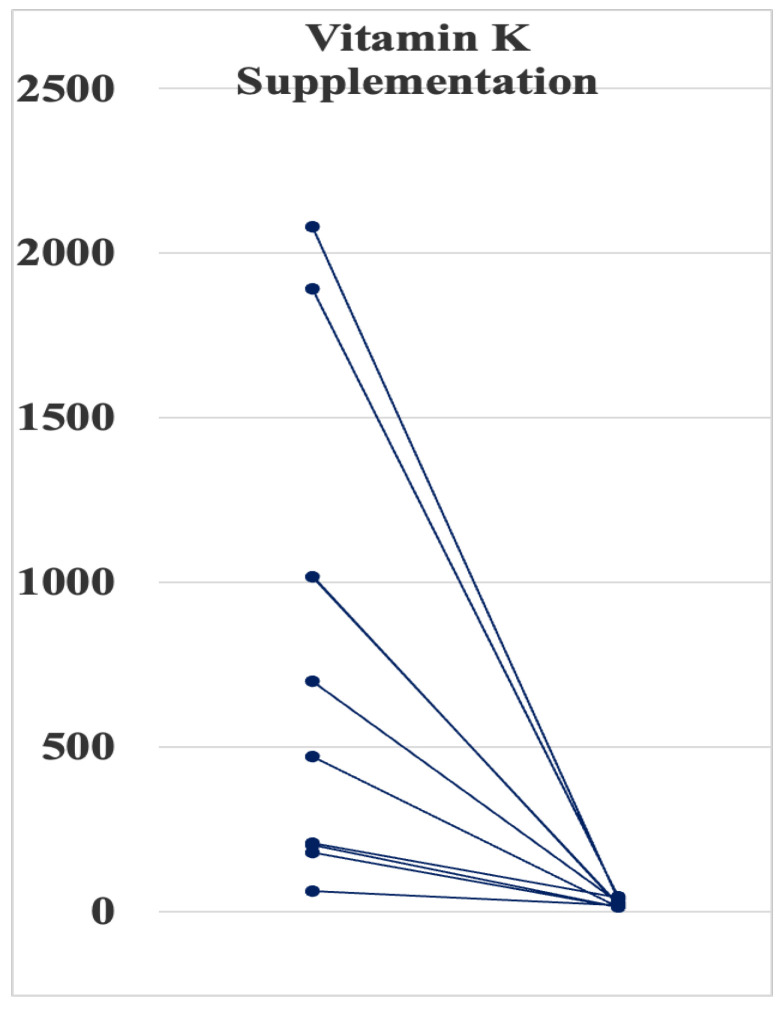
Effect of vitamin K2 supplementation for 3 months on Protein Induced by Vitamin K Absence or Antagonist-II levels in eight patients with severe motor and intellectual isabilities whose levels exceeded 100 mAU/mL.

**Table 1 nutrients-15-04525-t001:** Comparison of laboratory data between SMID patients with normal and elevated PIVKA-II.

	Elevated (*n* = 13)	Normal (*n* = 7)
PIVKA-II (mAU/mL) (Normal Range < 40)	616 ± 194 (44–2080)	28 ± 3 (21–39)
Blood coagulation test		
Prothrombin time (s) (normal range: 10.0–13.0)	13.9 ± 1.4	11.9 ± 0.3
PT (%) (normal range: 80–120)	78 ± 5	91 ± 5
PT-INR (normal range: 1.00–1.40)	1.25 ± 0.14	1.10 ± 0.05
aPTT (s) (normal range: 24.5–33.5)	35.9 ± 1.7	35.8 ± 1.6
Fibrinogen (mg/dL) (normal range: 200–400)	316 ± 27	332 ± 19
AT-III (%) (normal range: 5–125)	109 ± 4	113 ± 4
Duration of tube feeding (years)	10 ± 2	11.9 ± 0.3
Daily nutritional intake		
Vitamin K intake (µg/day)	53 ± 5	70 ± 9
Percentile of recommended daily vitamin K intake (%)	36.7 ± 2.6	46.5 ± 6.2
Energy intake (kcal/day)	950 ± 50	970 ± 102
Percentile of recommended daily energy intake (%)	43.0 ± 3.1	44.1 ± 5.3
Protein (g/day)	45 ± 3	45 ± 4
Fat (g/day)	29 ± 10	29 ± 4
Carbohydrate (g/day)	133 ± 10	127 ± 15
Nutritional parameters		
Albumin (g/dL)	3.6 ± 0.1 *	3.8 ± 0.2
Transferrin (mg/dL)	225 ± 10	220 ± 9
Transthyretin (mg/dL)	29 ± 2	31 ± 3
Retinol binding protein (mg/dL)	7.0 ± 0.9	6.4 ± 1.0
Ceruloplasmin (mg/dL)	32.7 ± 1.9	33.5 ± 2.6
Total cholesterol (mg/dL)	164 ± 7	175 ± 9
Triglyceride (mg/dL)	124 ± 21	79 ± 13
HDL cholesterol (mg/dL)	58 ± 6	66 ± 6
LDL cholesterol (mg/dL)	81 ± 6	92 ± 8
Cholinesterase (U/L)	312 ± 13	360 ± 19
Total lymphocyte count (/μL)	2003 ± 356	2064 ± 399
C-reactive protein (mg/dL)	0.56 ± 0.14	0.48 ± 0.17
Liver and kidney functions		
AST (IU/L)	23 ± 2	17 ± 2
ALT (IU/L)	21 ± 3	16 ± 1
γ-GTP (IU/L)	81 ± 14	56 ± 13
ALP (IU/mL)	136 ± 15	99 ± 8
LDH (U/L)	159 ± 11	145 ± 20
CPK (U/L)	57 ± 12	72 ± 20
Creatinine (mg/dL)	0.58 ± 0.16	0.56 ± 0.12
Blood urea nitrogen (mg/dL)	17.3 ± 6.2	17.4 ± 4.0

Data are expressed as mean ± standard deviation. * *p* < 0.05 vs. normal group. ALP: Alkaline Phosphatase, ALT: Alanine Aminotransferase, AST: Aspartate Aminotransferase, AT-III: Antithrombin III, aPTT: Activated Partial Thromboplastin Time, CPK: Creatine Phosphokinase, γ-GTP: Gamma-Glutamyl Transferase, HDL: High-Density Lipoprotein, LDH: Lactate Dehydrogenase, LDL: Low-Density Lipoprotein, PIVKA-II: Protein Induced by Vitamin K Absence or Antagonist-II, PT: Prothrombin Time, PT-INR: Prothrombin Time-International Normalized Ratio.

**Table 2 nutrients-15-04525-t002:** Comparison of laboratory data between patients with normal and elevated ucOC.

	Elevated (*n* = 9)	Normal (*n* = 11)
Osteocalcin (ng/mL)	30.2 ± 11.7	22.1 ± 9.9
ucOC (ng/mL) (normal range < 4.5 ng/mL)	7.59 ± 3.12	2.44 ± 1.08
Blood coagulation test		
Prothrombin time (s) (normal range: 10.0–13.0)	14.0 ± 6.4	12.2 ± 0.8
PT (%) (normal range: 80–120)	82 ± 25	86 ± 14
PT-INR (normal range: 1.00–1.40)	1.28 ± 0.64	1.08 ± 0.08
aPTT (s) (normal range: 24.5–33.5)	35.5 ± 7.9	35.1 ± 3.2
Fibrinogen (mg/dL) (normal range: 200–400)	344 ± 76	309 ± 93
AT-III (%) (normal range: 5–125)	113 ± 13	110 ± 12
Duration of tube feeding (years)	12 ± 8	13 ± 7
Daily nutritional intake		
Vitamin K intake (µg/day)	55 ± 25	61 ± 18
Percentile of recommended daily vitamin K intake (%)	39.0 ± 14.2	41.0 ± 12.2
Energy intake (kcal/day)	947 ± 205	970 ± 220
Protein (g/day)	48 ± 12	43 ± 12
Fat (g/day)	22 ± 12	36 ± 39
Carbohydrate (g/day)	136 ± 29	132 ± 37
Nutritional parameters		
Albumin (g/dL)	3.7 ± 0.4	3.6 ± 0.4
Transferrin (mg/dL)	221 ± 36	225 ± 31
Transthyretin (mg/dL)	31 ± 8	31 ± 3
Retinol binding protein (mg/dL)	7.4 ± 4.2	6.2 ± 1.9
Ceruloplasmin (mg/dL)	34.8 ± 8.0	30.8 ± 4.7
Total cholesterol (mg/dL)	157 ± 12	175 ± 9
Triglyceride (mg/dL)	136 ± 83	85 ± 36
HDL cholesterol (mg/dL)	55 ± 22	66 ± 14
LDL cholesterol (mg/dL)	75 ± 12	93 ± 24
Cholinesterase (U/L)	352 ± 62	310 ± 35
Total lymphocyte count (/μL)	2376 ± 1277	1737 ± 1010
C-reactive protein (mg/dL)	0.60 ± 0.60	0.47 ± 0.35
Liver and kidney functions		
AST (IU/L)	21 ± 10	21 ± 6
ALT (IU/L)	20 ± 10	19 ± 7
γ-GTP (IU/L)	72 ± 57	72 ± 35
ALP (IU/mL)	124 ± 67	112 ± 32
LDH (U/L)	150 ± 42	154 ± 42
CPK (U/L)	53 ± 48	59 ± 39
Creatinine (mg/dL)	0.64 ± 0.69	0.51 ± 0.28
Blood urea nitrogen (mg/dL)	18.7 ± 18.1	16.3 ± 13.6

Data are expressed as mean ± standard deviation. ALP: Alkaline Phosphatase, ALT: Alanine Aminotransferase, AST: Aspartate Aminotransferase, AT-III: Antithrombin III, aPTT: Activated Partial Thromboplastin Time, CPK: Creatine Phosphokinase, γ-GTP: Gamma-Glutamyl Transferase, HDL: High-Density Lipoprotein, LDH: Lactate Dehydrogenase, LDL: Low-Density Lipoprotein, PIVKA-II: Protein Induced by Vitamin K Absence or Antagonist-II, PT: Prothrombin Time, PT-INR: Prothrombin Time-International Normalized Ratio.

**Table 3 nutrients-15-04525-t003:** Changes of heel BMD T-score and Z-score at the calcaneus using quantitative ultrasonography during the observation period.

MK-4 Treatment	T-Score		Z-Score	
	Before	After	Before	After
Yes (*n* = 4)	−5.5 ± 1.0	−6.1 ± 0.9	−3.9 ± 1.3	−5.0 ± 2.1
No (*n* = 6)	−6.3 ± 1.0	−6.4 ± 0.8	−4.6 ± 1.2	−4.6 ± 1.1

MK: menaquinone.

## Data Availability

The datasets generated and/or analyzed during the current study are available from the corresponding on reasonable request.

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
