# Peer review of "The Effect of Vitamin K2 Supplementation on PIVKA-II Levels in Patients with Severe Motor and Intellectual Disabilities Undergoing Long-Term Tube Feeding"

_nutrients, 2023, doi:10.3390/nu15214525_

Round 1

Reviewer 1 Report

In their manuscript “Vitamin K deficiency in patients with severe motor and intellectual disabilities undergoing long-term tube feeding”, the authors assessed the K status in 20 patients with SMID in long term tube feeding and supplemented K” for 3 months to subjects with deficiency of that vitamin.

The main problem of this manuscript is that its title suggests that the authors will present the results of a prevalence study while the study is actually interventional and it provides an insufficiently detailed information on results of intervention (only the effect of Vitamine K” supplementation on PIVKA-II levels (figure 1) while a multivariate analyses is missing!). The title should be “The effects of…on PIVKAII levels” but the overall effects of supplementation should be studied better!

English language should be improved. Some sentences are incomprehensible (for example pg. 2 “The concept of SMID…is both original and administrative”?????).

There is a need for the order in the presentation of different sections of the manuscript. For example, in introduction, start with explaining the causes of SMID, explain why the subjects with SMID need frequently the artificial nutrition, characteristics of feeding formulas, deficiencies in SMID patients and K deficiency in particular, different risk factors for K deficiencies (among them feeding formulas) and why this deficiency is relevant (its consequences). Go step by step to make your work more comprehensive.

You should also add something about their nutritional status if you want to keep the comments on it in your discussion and I suggest saying something about the risks of K supplementation.

I do not think the table on page 4 is necessary while it would be better to describe the population (from 3.1. to 3.3.) in a table. Give the information on BMI instead of the weigh.

You have to explain why you supplemented from 2 to 3 capsules of the vitamin K.

You should add something on statistical analyses and put the ethics statement. Did you have the written consent?

Attention for your references. You should put the period after the brackets and not before [1,2,3].

Lines: 273 “due to the low prevalence”. Written in that way it seems that is an irrelevant problem to study.

Line 276: put in your results if you had some cases of sudden deaths and write something about that in the introduction.

The presentation in general needs to be improved (not so much the grammar or typing errors)

Author Response

Manuscript ID nutrients-2602365. R1

The effect of vitamin K2 supplementation on PIVKA-II levels in patients with severe motor and intellectual disabilities undergoing long-term tube feeding

Dear Editors and Reviewers:

Firstly, the authors wish to extend their gratitude to the editors and reviewers for their invaluable contributions and constructive suggestions. In response to the reviewers' insightful recommendations, we have substantially revised our manuscript and would like to resubmit this updated version. The suggestions provided were highly relevant and effectively addressed the shortcomings of the original manuscript. We have revised certain sections in accordance with the reviewers' comments and have duly corrected our manuscript.

Yours sincerely,

Hiromitsu Ohmori, MD., Ph.D.

Department of Pediatrics, National Hospital Organization, Yanai Medical Center,

95 Ihonosho, Yanai, Yamaguchi 742-1352, Japan.

PHONE: +81-820-27-0211

FAX: +81-820-27-1003

E-mail: h-h.ohmori@sound.ocn.ne.jp

Response to reviewers

Reviewer 1:

Comment 1. The title should be “The effects of … on PIVKA-II levels”.

We appreciate your suggestion. The title has been changed to "The effect of vitamin K2 supplementation on PIVKA-II levels in patients with severe motor and intellectual disabilities undergoing long-term tube feeding" (page 1, lines 2–3, red line).

Comment 2. The sentence of “The concept of SMID … is both original and administrative” is incomprehensive.

Thank you for your insightful comment. We have revised the sentence to clarify the disease background more effectively (page 2, lines 71–86, red line).

Comment 3. There is a need for the order in the preparation of different sections of the manuscript.

In response to your comment, we have revised the Introduction section to proceed more comprehensively, step by step (page 2, lines 72–87, red line). Furthermore, we have clarified the Discussion section regarding vitamin K1 and K2 supply (page 8, lines 299–338, red line).

Comment 4. You should add something about their nutritional status.

We have incorporated comments on biochemical nutritional parameters in the Results section (page 6, lines 217–219, red line). Furthermore, we have also added remarks concerning the risk of vitamin K supplementation, specifically thrombotic tendency, in the text (page 4, lines 138–140, red line).

Comment 5. It would be better to describe population (from supplemental Table 3 and 4) in a table. Give the information on BMI instead of the weigh.

In response to your suggestion, we have amalgamated Supplemental Tables 3 and 4, now presented as Supplemental Table 3 on page 4. Furthermore, we have incorporated BMI data into the Results section (page 5, lines 167–168, red line).

Comment 6. You have to explain why you supplemented from 2 to 3 capsules of the vitamin K2.

Thank you for your insightful suggestion. In Japan, the daily dosage and administration of MK-4 are established at 45 mg, administered thrice a day for adults. Moreover, as the safety of MK-4 has yet to be confirmed in pediatric subjects, we have reduced the daily dose to 30 mg in patients younger than 15 years old. This pharmacological information is elaborated upon at page 4, lines 131–140 (red line).

Comments 7. You should add something on statistical analyses and put the ethics statement.

We have incorporated multiple regression analysis into the Statistical Analyses section. Regarding the ethics statement, it is already present on page 4, lines 154–162.

Comment 8. You should put the period after the brackets and not before.

We apologize for the oversight in the formatting of our references. We have corrected the placement of the period to follow the closing bracket, as in "[1].", throughout the text.

Comment 9. Line 290 “due to the low prevalence”. Written in that way “it seems that is an irrelevant problem to study”.

Thank you for your guidance. We have amended the sentence to read, "it seems that this is an irrelevant problem to study" (page 9, line 350, red line).

Comment 10. Put in your results if you had some cases of sudden death and write something about in the Introduction.

We have noted that no patient experienced sudden death during the observation period (page 7, lines 249–250, red line). Furthermore, we have elaborated on the prevalence of cardiovascular or sudden death more precisely in the Discussion section (lines 353–355, red line).

Reviewer 2 Report

It is  a well written paper, PIVKA-II and uc-Oc have been frequently used together with ucMGP, Gas6 and other vitamin K-dependant proteins. Using ucOC and the ratio ucOC/OC to study "bone health" is complex, I think the authors should be more careful when they thread this path, especially in the discussion (their conclusion is to the point and actually is very good). Also being immobilised probably has more effects on bone structure/fracture risks than the nutritional defects. PIVKA-II is often decreased in many health scenarios. The clinical significance in patients with normal prothrombin time (PT) - sometimes adressed as a subclinical vitamin K deficiency is unclear in many situations, please adress in the Discussion. Vitamin K substitution with high doses of menaquinone (vitamnin K2-7 not menatetrione (vitamin K2-4 as in the present study) has only been shown to decrease media sclerosis in diabetic patients with uremia. The problem with this research field is too low dosages of vitamin K2 or vitamin K, too short observation times and not using clinical and MR/CT/sonography/tissue pathology evidence of improved vascular and tissue  health. There are many animal and cell experimental studies on vitamin K, most not relevant to humans. The type of patients in this study should have had ucGas6/Gas6 and TAM-receptor analyses added, as these are more relevant in neuroscience.

So in conclusion, a better balanced Discussion, reflecting the insequrity of clinical finds of PIVKA-II and ucOC. The stength of the paper is that the authors both revealed a vitamin K deficiency in this type of patients (nothing new), but tried and were successful to treat this. But most of the litterature fokus on menaquinones and not menatetrione and a paragraph on this should be added to the discussion. Also menaquinione supplementation is very expensive especially when used in higher dosages than recommended by FDA as compared to menatetriones. Synthetic alternatives to the soy-based (natto-based) may be less expensive. The nattobased vitamin K2 formulas  also contain an enzyme nattokinase, that have its own health effects.

Author Response

Reviewer 2:

Comment 1. The clinical significance in patients with normal prothrombin time (PT)- sometimes addressed as a subclinical vitamin K deficiency is unclear in many situations.

We have addressed the limitation of PT as a biomarker for vitamin K deficiency in the liver in the Discussion section (page 8, lines 299–304, red line).

Comment 2. The problem with this research field is too low dosages of vitamin K2 or vitamin K, too short observation times and not using clinical and MR/CT/sonography/tissue pathology evidence of improved.

As you highlighted, we administered 45 mg/day of MK-4 for only 3 months. Therefore, we have included comments on the long-term treatment of MK-4 in the Discussion section (page 9, lines 312–321). Moreover, we have incorporated heel BMD data obtained by ultrasonography for patients who could be measured both before and after the treatment (page 4, lines 142–145; page 8, lines 257–266; and Table 3, red line).

Comment 3. Most of the literature focus on menaquinones and not menatetrenone and a paragraph on this should be added to the discussion.

In response to your comment, we have clarified the role of vitamin K2 and K1 supplementation on bone and liver enzyme activity in the Discussion section (page 8, lines 301–336). Furthermore, we have commented on the potential for a natto-based formula in promoting bone health (page 8, lines 321–322, red line).